# MCMC-Correction of Score-Based Diffusion Models for Model Composition

## Abstract

Diffusion models can be parameterized in terms of either a score or an energy function. The energy parameterization is attractive as it enables sampling procedures such as Markov Chain Monte Carlo (MCMC) that incorporates a Metropolis–Hastings (MH) correction step based on energy differences between proposed samples. Such corrections can significantly improve sampling quality, particularly in the context of model composition, where pre-trained models are combined to generate samples from novel distributions. Score-based diffusion models, on the other hand, are more widely adopted and come with a rich ecosystem of pre-trained models. However, they do not, in general, define an underlying energy function, making MH-based sampling inapplicable. In this work, we address this limitation by retaining the score parameterization and introducing a novel MH-like acceptance rule based on line integration of the score function. This allows the reuse of existing diffusion models while still combining the reverse process with various MCMC techniques, viewed as an instance of annealed MCMC. Through experiments on synthetic and real-world data, we show that our MH-like samplers offer comparable improvements to those obtained with energy-based models, without requiring explicit energy parameterization.

## 1 Introduction

Significant advancements have recently been achieved in generative modelling across various domains (Brock et al., 2019; Brown et al., 2020; Ho et al., 2020). These models have become potent priors for a wide range of applications, including code generation (Li et al., 2022), text-to-image generation (Saharia et al., 2022), question-answering (Brown et al., 2020), and many others (Güngör et al., 2023; Wynn & Turmukhambetov, 2023). Among the generative models, diffusion models (Sohl-Dickstein et al., 2015; Song & Ermon, 2019; Ho et al., 2020) have arguably emerged as the most powerful class. Diffusion models learn to denoise corrupted inputs in small, gradual steps and are capable of generating samples from complex distributions. They have been successful in many domains, such as generating highly realistic images (Dhariwal & Nichol, 2021), modeling temporal point processes (Lüdke et al., 2023) and even generating neural network parameters (Wang et al., 2024).

Diffusion models also offer the capability of composed sampling, which combines pre-trained models to generate samples from a new distribution. This approach, known as model composition, has a rich history (Jacobs et al., 1991; Hinton, 2002; Mayraz & Hinton, 2000; Liu et al., 2022). For diffusion models, the most common form of composition is classifier-guided sampling, where the reverse process is augmented by a separate classifier model (Sohl-Dickstein et al., 2015; Dhariwal & Nichol, 2021; Ho & Salimans, 2021), but other compositions have also been explored (Du et al., 2023). The ability to compose new models without having to re-learn the individual components is especially appealing for diffusion models since their ever-increasing size and data hunger make them exceedingly costly to train (Aghajanyan et al., 2023). Therefore, developing sampling methods that work for pre-trained diffusion models is valuable.

The foundation of composed sampling for diffusion models is score-based, where we interpret diffusion models as predictors of the score function for the marginal distribution at each diffusion step (Song et al., 2021). From this perspective, MCMC methods, such as the Langevin algorithm (LA) (Roberts & Stramer, 2002) or

Hamiltonian Monte Carlo (HMC) sampling (Duane et al., 1987), emerge as viable options to incorporate. Augmenting the standard reverse process with additional MCMC sampling has been shown to improve composed sampling for diffusion models (Du et al., 2023; Song et al., 2021). However, we are restricted to unadjusted variants of these samplers, namely Unadjusted LA (U-LA) and Unadjusted HMC (U-HMC), which only require utilization of the score. This limitation means we cannot incorporate a Metropolis-Hastings (MH) correction step (Metropolis et al., 1953; Hastings, 1970), which requires evaluating the unnormalized density.

An intriguing alternative to directly modeling the score function is to model the marginal distribution with an energy function, from which the score can be obtained through explicit differentiation (Salimans & Ho, 2021; Song & Ermon, 2019). This parameterization connects diffusion models and energy-based models (EBMs) (LeCun et al., 2006) and offers several desirable properties. With an energy parameterization, we can evaluate the unnormalized density and guarantee a proper score function. This, in turn, enables an MH correction step when employing an MCMC-method, where the MH acceptance probability is computed from the energy function. Adding such a correction step has been shown to improve sampling performance in composed models (Du et al., 2023). Nevertheless, the score parameterization remains far more popular, as it avoids the direct computation of the gradient of the log density.

In this study, we build on the work in (Du et al., 2023) and introduce a novel approach to obtain an MH-like correction step directly from pre-trained diffusion models without relying on an energy-based parameterization. Specifically, we use a connection between the score and the energy to estimate the MH acceptance probability by approximating a line integral along the vector field generated by the score. This enables an improved sampling procedure for various pre-trained score-parameterized diffusion models. We find that our approximate method quantitatively results in improvements comparable to the energy parameterization without having to estimate the energy directly.

In summary, our main contributions are:

- We show that MH-like correction sampling can be directly applied to score-based models without requiring additional training.

- We introduce two efficient algorithms to approximate the energy difference used in MH and demonstrate that our pseudo-energy difference more accurately represents analytical energy differences than an explicitly trained energy model in a toy example while performing on par with the energy model on MNIST.

- We establish that the sampling accuracy improvements achieved with MCMC for energy-based models can also be attained for score-based models while offering superior runtime performance.

## 2  Background

### 2.1  Diffusion Models

We consider Gaussian diffusion models initially proposed by Sohl-Dickstein et al. (2015) and further improved by Song & Ermon (2019); Ho et al. (2020). Starting with a sample from the data distribution $x_0 \sim q(\cdot)$, we construct a Markov chain of latent variables $x_1, \ldots, x_T$ by iteratively introducing Gaussian noise to the sample $q(x_t|x_{t-1}) = \mathcal{N}\left(x_t; \sqrt{1 - \beta_t}x_{t-1}, \beta_t I\right)$, where $\beta_t \in [0, 1)$, $\forall t = 1, \ldots, T$ are known. For large enough $T$ we have $q(x_T) \approx \mathcal{N}(x_T; 0, I)$.

A diffusion model learns to gradually denoise samples by modeling the distribution of the previous sample in the chain $p_\theta(x_{t-1} \mid x_t), t = 1, \ldots, T$. Approximate samples from the data distribution $q(x_0)$ are obtained by starting from $x_T \sim \mathcal{N}(0, I)$ and sequentially sampling less noisy versions of the sample until the noise is removed. This is called the *reverse process*.

The reverse distribution is typically modeled as $p_\theta(x_{t-1}|x_t) = \mathcal{N}(x_{t-1}; \mu_\theta(x_t, t), \Sigma_\theta(x_t, t))$, since the posterior $q(x_{t-1}|x_t)$ can be well-approximated by a Gaussian distribution when the noise magnitude $\beta_t$ is sufficiently small. The mean is parameterized as $\mu_\theta(x_t, t) = \frac{1}{\sqrt{\alpha_t}}\left(x_t - \frac{\beta_t}{\sigma_t}\epsilon_\theta(x_t, t)\right)$, where $\alpha_t$ and $\sigma_t$ are positive and

defined by $\{\beta_t\}_{t=1}^T$ Ho et al. (2020). The noise prediction model $\epsilon_\theta(x_t, t)$, typically a neural network, is learned from data. We assume $\Sigma_\theta(x_t, t) = \beta_t I$ throughout unless otherwise stated.

## 2.2 Energy-based Models

Energy based-models (EBM) represent probability distributions with a scalar, non-negative energy function $E_\theta$, by assigning low energy to regions of the input space where the probability is high and high energy to regions where the distribution has little or no support:

$$p_\theta(x_t, t) = \frac{1}{Z_\theta(t)} \exp\left(-\frac{1}{\sigma_t} E_\theta(x_t, t)\right),$$
$$Z_\theta(t) = \int \exp\left(-\frac{1}{\sigma_t} E_\theta(x_t, t)\right) dx_t. \tag{1}$$

Here, we define $E_\theta$ as a time-dependent function and deliberately choose not to absorb $\sigma_t$ (introduced in the previous section) into $E_\theta$, to maintain a more explicit connection to diffusion models, as clarified in the next section. This time dependency can be seen as a sequence of energy functions, one for each diffusion step $t$. The normalization constant $Z_\theta$ is typically intractable, prohibiting computing a normalized density. However, $Z_\theta$ does not depend on the input $x_t$, making the so-called *score function* easy to compute:

$$\nabla_x \log p_\theta(x_t, t) = -\frac{1}{\sigma_t} \nabla_x E_\theta(x_t, t), \tag{2}$$

even though the gradient of the energy function can be costly to compute in practice.

## 2.3 Energy and Score Parameterized Diffusion Models

A popular method for training EBMs is denoising score matching (DSM). In DSM, assuming the data is perturbed by Gaussian noise, the loss function becomes identical to the one used for diffusion models (up to a factor of $\sigma_t^2$) (Song et al., 2021). This is achieved by identifying the noise prediction model, $\epsilon_\theta(x_t, t)$, as an EBM:

$$\epsilon_\theta(x_t, t) = \nabla_x E_\theta(x_t, t), \tag{3}$$

i.e., under the additional constraint that $\epsilon_\theta(x_t, t)$ defines a proper score. Thus, an EBM and a plain diffusion model only differ in their parameterization. We refer to the first as using an *energy parameterization* via $E_\theta$, while the second, since $\epsilon_\theta$ describes a pseudo-score, is referred to as using a *score parameterization*.

Both parameterizations have their advantages and disadvantages. The energy parameterization can evaluate the density $p_\theta(x_t, t)$ up to a normalization $Z_\theta(t)$, which enables various MCMC methods. Furthermore, by making the score equal to the gradient of an actual scalar function, we ensure a proper score. On the other hand, to evaluate the score function, $E_\theta$ must be explicitly differentiated, which can be costly.

The score parameterization is more flexible as it predicts an arbitrary vector field. While there is some empirical evidence that this improves sampling performance in diffusion processes (Du et al., 2023), this difference may primarily stem from model architecture (Salimans & Ho, 2021). Nevertheless, the score parameterization's direct estimation of the score function makes it more efficient for reverse process sampling and remains the more widely adopted approach. In the next section, we describe how these parameterizations affect the design of MCMC samplers for diffusion models.

## 2.4 MCMC Sampling For Diffusion Models

MCMC sampling is a promising strategy for improving diffusion model sampling since it can be combined with the reverse process. Just like the reverse process, there are MCMC methods which base their kernels on the score function, such as the Unadjusted Langevin Algorithm (U-LA) and the Unadjusted Hamiltonian Monte Carlo (U-HMC) (Roberts & Stramer, 2002; Duane et al., 1987; Neal et al., 1996). We let $\tau$ denote the index of the MCMC iterations (as opposed to $t$, which refers to the diffusion timestep).

For U-LA we use the kernel

$$k_t \left( x^{\tau+1} \mid x^\tau \right) \;=\; \mathcal{N} \left( x^{\tau+1}; \; x^\tau + \delta_t \nabla_x \log p_\theta(x^\tau, t), \; 2\delta_t I \right),$$

at diffusion step $t$, where $x^0 = x_t$, $\delta_t$ is the step size, and the chain is iterated for $L_t$ steps.

For U-HMC we augment the state with momenta $\mathbf{v}^\tau \sim \mathcal{N}(0, M_t)$ (diagonal mass $M_t$), and propose a new state by applying $\ell_t$ leapfrog steps of size $\varepsilon_t$ under the same score field. Writing

$$(x^{\tau+1}, \mathbf{v}^{\tau+1}) \;=\; \mathrm{LF}_{\ell_t, \varepsilon_t}(x^\tau, \mathbf{v}^\tau),$$

for the leapfrog map (which in our case is defined with the score $\nabla_x \log p_\theta(x, t)$ as potential gradient), the kernel for $x$ is given implicitly by this deterministic proposal after marginalizing out $\mathbf{v}$. In practice, we may perform $L_t$ such proposals per diffusion step $t$.

These methods are called unadjusted since, as $L_t$ grows, the Markov chains will converge to the target distribution only in the limit of infinitesimal step sizes. By adding, for instance, a Metropolis–Hastings (MH) correction step, we can sample with larger step sizes and still converge to the target distribution (Metropolis et al., 1953; Hastings, 1970). With the correction, we sample a candidate $\hat{x} \sim k_t(\cdot \mid x^\tau)$ and accept it as the new iterate with probability

$$\alpha \;=\; \min\left( 1, \; \frac{p_\theta(\hat{x}, t)}{p_\theta(x^\tau, t)} \, \frac{k_t\left(x^\tau \mid \hat{x}\right)}{k_t\left(\hat{x} \mid x^\tau\right)} \right), \tag{4}$$

so that $x^{\tau+1} = \hat{x}$ with probability $\alpha$, and $x^{\tau+1} = x^\tau$ otherwise. With an MH correction, U-LA becomes LA, and U-HMC becomes HMC.

The model $p_\theta$ appears only through a ratio, so a normalized density is not required. When $p_\theta$ is parameterized as an EBM (see (1)), the ratio simplifies to

$$\frac{p_\theta(\hat{x}, t)}{p_\theta(x^\tau, t)} \;=\; \exp\left( \frac{1}{\sigma_t} \left( E_\theta(x^\tau, t) - E_\theta(\hat{x}, t) \right) \right), \tag{5}$$

which allows us to directly evaluate the MH acceptance probability, making it straightforward to construct an adjusted MCMC sampler. This offers a key advantage over the score parameterization, where only an approximation of the score is accessible which cannot directly be used to compute the probability ratio needed in MH.

## 2.5 Sampling from Composed Models

Composed sampling is a powerful feature of diffusion models that enables sampling from new target distributions by combining multiple pre-trained models. Rather than retraining a model for every new task or data combination, one can reuse existing components. This flexibility is especially appealing in large-scale settings, where retraining is often prohibitively expensive.

The most common form of composition is *guidance* (Dhariwal & Nichol, 2021), where the goal is to sample from a distribution conditioned on a class label $y$,

$$q(x_0 \mid y) \propto q(x_0) q(y \mid x_0). \tag{6}$$

This is implemented by modifying the score function at each diffusion step as

$$\nabla_x \log p_\theta(x_t, t) + \lambda \nabla_x \log p_\varphi(y \mid x_t, t), \tag{7}$$

where $p_\theta$ is an unconditional diffusion model and $p_\varphi$ is a classifier predicting class $y$. A hyperparameter $\lambda$ controls the strength of the conditioning. We refer to this approach as *classifier-full guidance*. Other variants include reconstruction guidance (Chung et al., 2023; Ho et al., 2022) and classifier-free guidance (Ho & Salimans, 2021).

More generally, Du et al. (2023) explore a range of composition types beyond guidance, including *products*, *negations*, and *mixtures*. A product composition—of which guidance can be seen as a special case—is defined as

$$q^\Pi(x_0) \propto \prod_i q^i(x_0), \tag{8}$$

and leads to the composed model at diffusion step $t$,

$$p_\theta^\Pi(x_t, t) \propto \prod_i p_{\theta_i}^i(x_t, t) = \exp\left(-\frac{1}{\sigma_t} \sum_i E_{\theta_i}^i(x_t, t)\right). \tag{9}$$

This distribution is then used as the target in MCMC sampling, resulting in improved sampling performance.

Importantly, the factorization in (8) only strictly holds at $t = 0$; at intermediate diffusion steps, the composed model $p_\theta^\Pi(x_t, t)$ does not generally correspond to the true marginal of any product data distribution Du et al. (2023). This becomes problematic when relying solely on the reverse process, which assumes access to a valid score function for the true intermediate marginals. However, this construction remains valid and effective from the perspective of *annealed MCMC* Neal (2001), where the overall sampling procedure is interpreted as a chain targeting a sequence of gradually evolving distributions. From this viewpoint, the intermediate distributions $p_\theta^\Pi(x_t, t)$ are treated as design choices that guide the chain toward the final target $q^\Pi(x_0)$, and asymptotic correctness is still preserved. In practice, since diffusion models are trained using denoising score matching, the sampling process converges to a denoised version of $q^\Pi(x_0)$, which can be made arbitrarily close to the true distribution by construction.

Note for models using a score-based parameterization, a pseudo-score for this type of composition is equal to $-\frac{1}{\sigma_t} \sum_i \epsilon_{\theta_i}^i(x_t, t)$.

## 3 MCMC Correction Step For Score Parameterization

We propose combining the energy parameterization properties with the performance and practical accessibility of the score parameterization. Instead of using an energy parameterization and computing the score by differentiation, we take the complementary approach: using a score parameterization and computing the change in (pseudo-)energy by integrating the score.

### 3.1 Pseudo-energy Difference and MH-like Correction

This section describes how MCMC acceptance probabilities can be approximated given a score function. The MH acceptance probability in (4) is based on the relative probability of the new candidate $\hat{x}$ and the current sample $x^\tau$. The transition probabilities given by the kernel $k_t(\cdot \mid \cdot)$ are assumed to be simple to compute, and we focus on the quotient $p_\theta(\hat{x}, t)/p_\theta(x^\tau, t)$. To compute the MH acceptance probability $\alpha$, we only need to evaluate the unnormalized target distribution. For an EBM, this can be expressed in terms of the difference in energy at $\hat{x}$ and $x^\tau$, see (5). That is, we do not need to compute the absolute value of the energy, only the difference.

To express the acceptance probability in terms of the score function of an EBM, we write the difference in energy as a line integral over a curve $\mathcal{C}$

$$E_\theta(x^\tau, t) - E_\theta(\hat{x}, t) = -\int_\mathcal{C} \nabla_r E_\theta(r, t) \cdot dr = -\int_0^1 \nabla_r E_\theta(r(s), t) \cdot r'(s)\, ds, \tag{10}$$

where $r(s)$ is a parameterization of $\mathcal{C}$ such that $r(0) = x^\tau$ and $r(1) = \hat{x}$. The choice of curve is arbitrary (under mild conditions), since $E_\theta$ is a scalar field.

For a score-parametrized diffusion model, we propose using a similar approach and calculating an MH-like ratio as follows:

$$\alpha = \min\left(1, \exp\left[\frac{1}{\sigma_t} f(\hat{x}, x^\tau, t)\right] \frac{k_t(x^\tau \mid \hat{x})}{k_t(\hat{x} \mid x^\tau)}\right), \tag{11}$$

where

$$f(\hat{x}, x^\tau, t) = -\int_0^1 \epsilon_\theta(r(s), t) \cdot r'(s) \, \mathrm{d}s, \tag{12}$$

representing our constructed *pseudo-energy difference*. This expression can be seen as integrating the vector field $\epsilon_\theta$ along a path from $x^\tau$ to $\hat{x}$, thereby approximating the change in a scalar potential—if such a potential existed. Note that if $\epsilon_\theta(x, t) = \nabla_x F(x, t)$ for some function $F$, (12) can be interpreted as recovering an (unknown) energy function, and in this case (11) agrees with (4). In general, however, no such function $F$ exists, and the expression (12) depends on the path $r$ that is integrated over. Nevertheless, we propose using (11) to directly model an MH-like acceptance probability to be used in an MCMC sampling scheme.

Since (12) in general depends on the path $r$ between $x^\tau$ and $\hat{x}$, we propose two variants for the curve $\mathcal{C}$: a straight line connecting the two points, and an HMC–curved path that follows the leapfrog trajectory of the proposal. Both variants are approximated with the trapezoidal rule, with the number of subsegments treated as a hyperparameter. In the straight-line case, only the two endpoints can be reused from the MCMC proposal and all intermediate scores must be evaluated. In the curved-path case, many of the required scores are already computed by the HMC kernel, and thus can be reused without extra evaluations. By aligning the integration path with the proposal trajectory, we achieve higher numerical accuracy without incurring additional model evaluations.

Note that estimating the line integral requires evaluating $\epsilon_\theta$ at internal points of $\mathcal{C}$, which incurs extra computation (except when values are reused in the HMC case). However, this approach avoids differentiating through the model: we estimate the score function directly using $\epsilon_\theta$, whereas in the energy parameterization one only evaluates the energy at $x^\tau$ and $\hat{x}$, but must differentiate $E_\theta$ to obtain the score.

An overview of the full sampling procedure is provided in Algorithm 1. At each diffusion step, an optional reverse update is followed by an MCMC refinement targeting the intermediate distribution, with the pseudo-energy difference estimated either by Algorithm 2 (straight line) or Algorithm 3 (curved path). This formulation aligns with the annealed MCMC framework, where both the reverse step and the MCMC kernel act as design choices guiding the chain toward the final distribution. Including the reverse step typically improves sample quality (Du et al., 2023).

### 3.2 MH-correction for Composition Models

The pseudo-energy difference for compositions can be derived based on their specific definitions. Our proposed method applies directly to product compositions. We calculate a pseudo-energy difference, corresponding to $E_\theta^\Pi(x^\tau, t) - E_\theta^\Pi(\hat{x}, t))$ for an EBM (defined in (9)), as

$$-\int_0^1 \sum_i \epsilon_{\theta_i}^i(r(s), t) \cdot r'(s) \, \mathrm{d}s. \tag{13}$$

Guidance is a specific case of product composition, where the pseudo-score is composed of two terms according to (7): the unconditional diffusion model $\epsilon_\theta(x_t, t)$ and the score of a classifier $p_\varphi(y \mid x_t, t)$. Since $p_\varphi(y \mid x_t, t)$ can be evaluated directly, only the pseudo-energy difference for $\epsilon_\theta(x_t, t)$ requires computation using the line integral in (13).

The pseudo-energy difference for a negation composition (as defined in (Du et al., 2023)) can be computed analogously to products, as negations follow a similar additive structure in their pseudo-scores.

Mixture compositions (as defined in (Du et al., 2023)), on the other hand, cannot be expressed as a pseudo-energy difference, since mixtures do not naturally conform to an additive structure analogous to products or negations. However, mixtures can be addressed by first sampling a component distribution according to the mixture definition and then generating a sample from that distribution. The MH-correction can subsequently be applied to this sampled distribution, providing a seamless way to handle mixture compositions within our framework.

This generalization allows our method to support advanced use cases such as classifier guidance, multi-modal fusion, and spatially structured prompts, without requiring retraining or access to energy-based models.

---

**Algorithm 1** Annealed MCMC with MH-like correction

---

**Require:** Score function $\epsilon_\theta(\cdot, t)$; reverse-diffusion schedule $(\alpha_t, \beta_t, \sigma_t)_{t=1}^T$; kernel family $\{k_t(\cdot \mid \cdot)\}_{t=1}^T$ (LA or HMC) with kernel hyperparameters; MCMC steps per diffusion step $\{L_t\}$; integration mode $\texttt{mode} \in \{\texttt{line}, \texttt{curve}\}$; line segments $n$ (if $\texttt{line}$); per-leapfrog subsegments $m$ (if $\texttt{curve}$).

1: Sample initial $x_T \sim \mathcal{N}(0, I)$.
2: **for** $t = T$ **to** 1 **do**
3:     *(Optional) Reverse step:* update $x_{t-1}$ from $x_t$ using the standard reverse-diffusion update at time $t$
4:     **if** $t > 1$ **then**
5:         $t' \leftarrow t - 1$              ▷ all MCMC quantities live at time $t'$
6:         Initialize MCMC: $x^0 \leftarrow x_{t'}$.
7:         **for** $\tau = 1$ **to** $L_t$ **do**
8:             Propose candidate $\hat{x} \sim k_{t'}(\cdot \mid x^{\tau-1})$        ▷ LA or HMC kernel
9:             $f \leftarrow \texttt{EstimatePseudoEnergy}(x^{\tau-1}, \hat{x}, t'; \texttt{mode}, n, m)$    ▷ Alg. 2 or Alg. 3
10:            Compute acceptance

$$\alpha = \min\left(1, \, \exp\left(\tfrac{1}{\sigma_{t'}} f\right) \cdot \frac{k_{t'}(x^{\tau-1} \mid \hat{x})}{k_{t'}(\hat{x} \mid x^{\tau-1})}\right)$$

11:            With probability $\alpha$: $x^\tau \leftarrow \hat{x}$; otherwise $x^\tau \leftarrow x^{\tau-1}$.
12:         **end for**
13:         Set $x_{t'} \leftarrow x^{L_t}$.
14:     **end if**
15: **end for**
16: **return** $x_0$

---

**Algorithm 2** EstimatePseudoEnergy (straight-line path)

---

**Require:** Current $x$, candidate $\hat{x}$, time $t$, number of segments $n \geq 2$; access to $\epsilon_\theta(\cdot, t)$. Note that $\epsilon_\theta(x, t)$ and $\epsilon_\theta(\hat{x}, t)$ are available from cache (Algorithm 1). The routine may reuse them instead of re-evaluating

1: $\Delta r \leftarrow \frac{1}{n-1}(\hat{x} - x)$
2: $r_{\text{prev}} \leftarrow x$;    $g_{\text{prev}} \leftarrow \epsilon_\theta(x, t)$;    $f \leftarrow 0$
3: **for** $j = 1$ **to** $n - 1$ **do**
4:     $r_j \leftarrow r_{\text{prev}} + \Delta r$
5:     $g_j \leftarrow \epsilon_\theta(r_j, t)$                ▷ reuse cached value at endpoints
6:     $f \leftarrow f - \frac{1}{2}\left(g_{\text{prev}} + g_j\right) \cdot \Delta r$
7:     $r_{\text{prev}} \leftarrow r_j$;    $g_{\text{prev}} \leftarrow g_j$
8: **end for**
9: **return** $f$

*Complexity (extra score evaluations).* If endpoint scores are reused, this routine performs $n - 2$ new $\epsilon_\theta(\cdot, t)$ calls; otherwise $n$ calls.

## 4 Results

In this section, we present an empirical evaluation of our MH-like correction method, examining both the accuracy of the pseudo-energy differences and the quality of the generated samples. The experiments are designed to span a spectrum of difficulty: from controlled, low-dimensional setups where models can be trained from scratch and analytical solutions are available, to more realistic high-dimensional scenarios involving pre-trained models. Our two primary objectives are (1) to compare our proposed approach against a true energy parameterization when available, and (2) to assess the sampling improvements achieved over the standard reverse process when augmented with MCMC steps.

The experiments in Sections 4.1, 4.2, and the first part of 4.3 involve training diffusion models using both energy and score parameterizations. The score parameterization follows a noise prediction model, $\epsilon_\theta(x_t, t)$,

---

**Algorithm 3** EstimatePseudoEnergy (HMC-curved path)

---

**Require:** Current $x$, candidate $\hat{x}$, time $t$; leapfrog trajectory $\{x^{(i)}\}_{i=0}^{\ell_t}$ with $x^{(0)}{=}x$, $x^{(\ell_t)}{=}\hat{x}$; subsegments per leapfrog step $m \geq 2$; access to $\epsilon_\theta(\cdot, t)$. Note that $\{\epsilon_\theta(x^{(i)}, t)\}_{i=0}^{\ell_t}$ are available from cache (Algorithm 1) and may be reused instead of re-evaluating.

1: $f \leftarrow 0$
2: **for** $i = 1$ **to** $\ell_t$ **do**                           ▷ integrate along each LF segment $x^{(i-1)} \rightarrow x^{(i)}$
3:      $r_{\text{prev}} \leftarrow x^{(i-1)}; \quad g_{\text{prev}} \leftarrow \epsilon_\theta(x^{(i-1)}, t)$
4:      $\Delta r_i \leftarrow \frac{1}{m-1}\big(x^{(i)} - x^{(i-1)}\big)$
5:      **for** $j = 1$ **to** $m - 1$ **do**
6:          $r_{i,j} \leftarrow r_{\text{prev}} + \Delta r_i$
7:          $g_{i,j} \leftarrow \epsilon_\theta(r_{i,j}, t)$                  ▷ reuse cached leapfrog scores if available
8:          $f \leftarrow f - \frac{1}{2}\big(g_{\text{prev}} + g_{i,j}\big) \cdot \big(r_{i,j} - r_{\text{prev}}\big)$
9:          $r_{\text{prev}} \leftarrow r_{i,j}; \quad g_{\text{prev}} \leftarrow g_{i,j}$
10:     **end for**
11: **end for**
12: **return** $f$

---

*Complexity.* With cached leapfrog scores, this routine performs $\ell_t\,(m-2)$ new $\epsilon_\theta(\cdot, t)$ calls (one per interior subsegment); without caching it performs $\ell_t\,(m-1)$ calls. Setting $m{=}2$ reduces to a trapezoid rule that uses only leapfrog endpoints.

---

while the energy parameterization defines an energy function as $E_\theta(x_t, t) = \|x_t - s_\theta(x_t, t)\|_2^2$, as in (Du et al., 2023). We use identical network architectures for $\epsilon_\theta$ and $s_\theta$. Both models are trained with the standard diffusion loss (Ho et al., 2020), with the energy model's score function obtained through explicit differentiation.

The later experiments utilize only pre-trained score-based diffusion models, as pre-trained energy-based models are unavailable for direct comparison. We evaluate both unadjusted and MH-corrected versions of Langevin and Hamiltonian Monte Carlo, comparing them against the standard reverse process, which serves as the baseline.

For the MH-like correction, we examine two types of integration paths: line and curve. The line follows a direct path between $x^\tau$ and $\hat{x}$, while the curve integrates along the trajectory formed by HMC leapfrog steps. The number of points for the trapezoidal rule's mesh is treated as a hyperparameter. Since points like $x^\tau$ and $\hat{x}$ are already included, the hyperparameter refers to the additional points, which are evenly distributed along the curve.

Complete training details, hyperparameter settings, and implementation specifics are deferred to Appendix A.

### 4.1   Evaluating Pseudo-Energy Differences

To evaluate the accuracy of pseudo-energy differences, we conducted experiments on a synthetic 2D dataset, generated from a bivariate Gaussian distribution to allow access to analytical solutions, and a higher-dimensional dataset, MNIST (Deng, 2012). For each experiment, we trained 10 score and energy models independently from scratch. For evaluation, we sampled 2k pairs of points $(x_t^1, x_t^2)$ via the forward process at various diffusion steps $t$, and these pairs were used to compute the (pseudo-)energy difference $\Delta E$ for both the score and energy models (and analytically when available). The pseudo-energy difference was computed along a linear curve connecting the two points, using five points for numerical integration.

**2D Gaussian:** For the 2D Gaussian dataset, the relative error metric is defined as $|\Delta E_{\text{pred}} - \Delta E_{\text{true}}|/|\Delta E_{\text{true}}|$, where $\Delta E_{\text{pred}}$ is the predicted energy difference and $\Delta E_{\text{true}}$ is the analytical energy difference. The median relative error was calculated across all sampled pairs for each trained model, and the mean and standard deviation of this metric were computed across the 10 models. Interestingly, the score model achieved a lower relative error $0.071 \pm 0.005$ compared to the energy model $0.084 \pm 0.004$, demonstrating better alignment with the true energy differences.

**MNIST:** For the MNIST dataset, where analytical energy differences are unavailable, we used a symmetric relative error metric defined as $2|\Delta E_{\text{score}} - \Delta E_{\text{energy}}|/(|\Delta E_{\text{score}}| + |\Delta E_{\text{energy}}|)$. The median relative error was calculated across all sampled pairs for each trained model, and the mean and standard deviation were computed across the 10 models. This yielded a mean relative error of $0.030 \pm 0.002$ indicating that the energy differences predicted by the score and energy models align closely, even in this higher-dimensional setting.

### 4.2 2D Composition

To investigate the effectiveness of our MH-like correction in a controlled yet expressive setting, we replicate the 2D composition experiment introduced by Du et al. (2023), using their publicly available codebase[1] as a foundation. Our experimental setup mirrors theirs unless otherwise specified.

A 2D density pair is composed via multiplication into a complex distribution, as in (9): a Gaussian mixture with 8 modes in a circle and a uniform distribution covering two of the modes. For a visual representation of the two individual distributions and their resulting product distribution together with samples from the reverse diffusion and HMC corrected samples, see Figure 1. The baseline reverse diffusion process uses $T = 100$ steps. In the MCMC variants, following Du et al. (2023), we omit the optional reverse step for a fair comparison. MCMC sampling runs for $L_t = 10$ at each $t$, with (U-)HMC using 3 leapfrog steps per MCMC step. We evaluate performance using three metrics. The first is negative log-likelihood (NLL), which

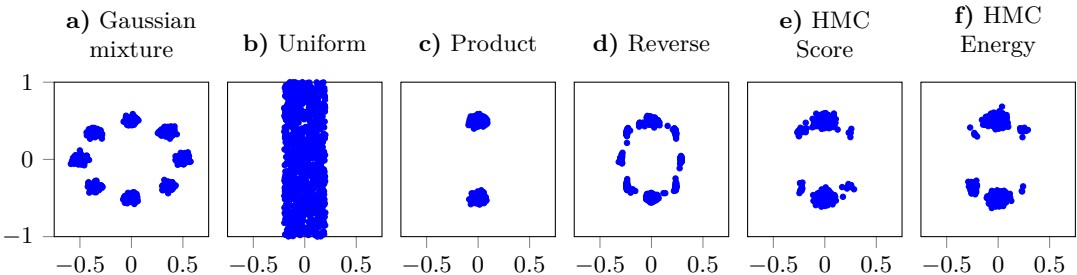

Figure 1: Samples from: **a-b):** the component distributions: a Gaussian mixture and a uniform distribution, **c):** the true product distribution, **d):** a standard score parameterized reverse process, **e-f):** HMC sampling using score and energy parameterization respectively.

assesses the likelihood of generated samples under the true data distribution. To address potential samples outside the true distribution's support, we extend it by adding a small uniform probability. The second metric is a Gaussian mixture model (GMM), where we fit bi-modal GMMs to samples from both the true and model distributions and compute the Frobenius norm of the variance mean difference. Finally, we use the Wasserstein-2 distance ($W_2$) to measure the discrepancy between the data and model distributions by computing the optimal assignment between sampled sets (Villani, 2009).

We present quantitative results for the 2D composition in Table 1(a), averaged over 10 independent trials. In each trial, we train the diffusion models from scratch and sample 2000 points using different MCMC methods. The results show that the corrected sampling methods outperform the unadjusted ones. HMC variants yield better results than Langevin, while the reverse process performs worse. Score and energy parameterizations exhibit similar NLL and GMM performance within their respective methods. However, with HMC, the score parameterization significantly outperforms the energy parameterization in $W_2$. Performance also saturates with as few as three points in the trapezoidal rule.

Additionally, we measured memory usage and runtime during this experiment, see Table 1(b). The experiment was implemented in JAX and run on a desktop computer equipped with an NVIDIA GeForce RTX 3060 GPU. Score-based parameterization was more than twice as memory-efficient as energy-based parameterization and, with the exception of LA with 8 extra trapezoidal evaluations, faster for the corresponding MCMC methods. Notably, the HMC curve variant was significantly faster. While our approach requires more model

---

[1]https://github.com/yilundu/reduce_reuse_recycle

evaluations, this does not necessarily make it slower or more memory-intensive than using an energy-based model. However, these results are implementation-dependent, and further investigation is needed to confirm whether these trends generalize to other setups.

Table 1: Quantitative results for different samplers in the 2D composition experiment. (a) shows performance metrics (NLL, GMM, and $W_2$) based on 10 independent trials, with lower values indicating better performance. (b) reports average runtime (in seconds) and peak memory consumption (in MiB). For the score parameterization, we include variants with different numbers of additional points in the trapezoidal rule (e.g., 1L, 3L, 8L) and different integration paths ("L" for a straight line and "C" for the HMC trajectory).

**(a) Performance metrics**

| | Sampler | NLL↓ | GMM↓ | $W_2$↓ |
|---|---|---|---|---|
| Energy | Reverse | 8.22(0.21) | 27.01(1.34) | 5.81(0.19) |
| | U-LA | 7.52(0.22) | 14.61(1.35) | 4.19(0.45) |
| | LA | 6.50(0.30) | 14.66(1.46) | 4.24(0.55) |
| | U-HMC | 5.72(0.18) | 6.53(0.91) | 4.19(1.25) |
| | HMC | **4.09(0.14)** | **3.33(0.65)** | **4.12(1.44)** |
| Score | Reverse | 8.15(0.24) | 26.88(1.20) | 5.80(0.20) |
| | U-LA | 7.57(0.12) | 14.99(0.62) | 4.44(0.63) |
| | LA-3L | 6.45(0.20) | 14.28(1.07) | 4.03(0.52) |
| | LA-5L | 6.61(0.17) | 15.19(0.92) | 4.22(0.46) |
| | LA-10L | 6.53(0.17) | 14.75(0.91) | 4.20(0.51) |
| | U-HMC | 5.77(0.12) | 6.90(0.71) | 3.39(0.77) |
| | HMC-3L | 4.29(0.13) | 3.72(0.61) | 2.92(1.02) |
| | HMC-5L | **4.07(0.13)** | 3.08(0.69) | **2.68(1.20)** |
| | HMC-10L | **4.07(0.14)** | 3.17(0.56) | 2.87(0.89) |
| | HMC-C | **4.07(0.12)** | **3.06(0.54)** | 2.94(0.90) |

**(b) Runtime and memory usage**

| | Sampler | Time | Memory |
|---|---|---|---|
| Energy | Reverse | 0.22(0.00) | 5252 |
| | U-LA | 1.54(0.01) | 5252 |
| | LA | 9.13(0.08) | 5252 |
| | U-HMC | 2.36(0.05) | 5254 |
| | HMC | 21.02(0.06) | 5256 |
| Score | Reverse | 0.11(0.00) | 2178 |
| | U-LA | 0.93(0.01) | 2180 |
| | LA-3L | 4.77(0.13) | 2180 |
| | LA-5L | 7.60(0.02) | 2180 |
| | LA-10L | 10.53(0.15) | 2180 |
| | U-HMC | 1.19(0.01) | 2180 |
| | HMC-3L | 7.08(0.02) | 2180 |
| | HMC-5L | 9.56(0.01) | 2180 |
| | HMC-10L | 10.64(0.01) | 2180 |
| | HMC-C | 1.48(0.04) | 2180 |

As also observed by Du et al. (2023), the reverse sampler corresponds to product composition obtained by directly adding score functions, which therefore performs poorly in this setting. This highlights the motivation for annealed MCMC: by treating intermediate distributions as design choices that guide the chain toward the target, annealing achieves improved results. This explains the large gap between the reverse method and the annealed MCMC variants reported in Table 1.

## 4.3 Guided Diffusion

We evaluate our proposed sampling methods for guided diffusion on the CIFAR-100 (Krizhevsky & Hinton, 2009) and ImageNet (Deng et al., 2009) datasets. The sampling process is based on a score function defined in (7). For both datasets, the marginal score, $\nabla_x \log q(x_t)$, is estimated using an unconditional diffusion model parameterized by a UNet architecture. For the guidance model, we use classifier-full guidance, training a time-dependent classifier to predict class labels across all diffusion steps, $p_\varphi(y \mid x_t, t)$. This classifier shares its architecture with the encoder part of the UNet used for the diffusion model and is extended with a dense output layer. The guidance scale is set to $\lambda = 20.0$ across all experiments. Sampling is based on the standard reverse process with $T = 1000$, and additional MCMC steps are incorporated to refine the generated samples.

To quantify generation quality, we use three evaluation metrics: the Fréchet Inception Distance (FID) (Heusel et al., 2017), which compares the distribution of generated and real images; classification accuracy, based on a separate pre-trained classifier applied to generated samples; and, for ImageNet, an additional top-5 accuracy metric.

**CIFAR-100:** For CIFAR-100, we trained the diffusion models from scratch using the same UNet architecture and training settings as in Ho et al. (2020), which were originally designed for CIFAR-10 (Krizhevsky & Hinton, 2009). The MCMC samplers add $L_t = 2$ or 6 extra MCMC steps at each diffusion step $t$ for (U-)HMC and (U-)LA, respectively, with (U-)HMC using three leapfrog steps per MCMC step.

Table 2: Accuracy and FID score for classifier-full guidance on CIFAR-100. The metrics are based on 50k generated samples for each sampling method with both energy and score models.

|        | Sampler | Accuracy [%]↑ | FID↓ |
|--------|---------|---------------|------|
|        | Reverse | 72.6          | 33.4 |
|        | U-LA    | **87.3**      | 24.6 |
| Energy | LA      | 80.0          | 12.7 |
|        | U-HMC   | 87.2          | 25.4 |
|        | HMC     | 84.9          | **12.4** |
|        | Reverse | 74.2          | 31.8 |
|        | U-LA    | **82.9**      | 25.9 |
| Score  | LA-10L  | 75.2          | 15.5 |
|        | U-HMC   | 79.0          | 28.6 |
|        | HMC-3C  | 75.8          | **13.3** |

For this experiment, more points are needed in the trapezoidal rule's mesh than in the 2D experiment. Based on previous insights, for HMC we integrate only along the curve from the leapfrog steps, with an additional midpoint evaluation, resulting in three extra model evaluations per HMC step. For LA, we use ten points along the line, resulting in eight extra evaluations per step.

Recognizing the impact of the step length on MCMC methods in general, we parameterize the step length as a function of the beta-schedule $\delta_t = a\beta_t^b$. We conducted a simple parameter search for parameters $a$ and $b$, to determine a suitable step length for each MCMC variant.

The results are shown in Table 2. Average accuracy is obtained using a separate classifier trained exclusively on noise-free pairs $(x_0, y)$, following the VGG-13-BN architecture (Simonyan & Zisserman, 2014). The table shows a general trend of improvement over the baseline reverse process when additional MCMC steps are added. In particular, the MH-corrected samplers LA and HMC show significant improvements in FID scores, which are arguably the more important metric for image generation.

Comparing the score and energy parameterizations, their performances share similar characteristics. Interestingly, the reverse process favors the score parameterization, supporting the claim that this less restricted approach better models the score function. However, the energy parameterization sees larger improvements from the added MCMC steps. This indicates, perhaps, that direct energy estimation provides a better correction step compared to our method of approximating the pseudo-energy difference from $\epsilon_\theta$. Although the energy-based method performs slightly better in this setting, our MH-corrected sampling methods achieve comparable improvements without requiring an energy model.

**ImageNet:** For ImageNet, training diffusion models from scratch is computationally expensive, so we rely on pre-trained models. Score-based models are publicly available through the OpenAI GitHub repository[2], as provided by Dhariwal & Nichol (2021). Unfortunately, no equivalent pre-trained energy-based models are available. Given the high computational demands of large-scale diffusion models, we focus solely on evaluating HMC and compare it to the reverse process. The HMC sampler adds $L_t = 2$ MCMC steps per diffusion step $t$, with each step consisting of three leapfrog steps. For the trapezoidal rule, we incorporate the points from the leapfrog steps and add two additional points between each leapfrog step. The step length parameterization and tuning follow the same procedure as in CIFAR-100.

The results can be seen in Table 3. Accuracy metrics are computed using a pre-trained RegNetX-8.0GF Radosavovic et al. (2020) classifier. The reverse process and HMC perform very similarly in average accuracy, but our method shows a slight improvement in top-5 average accuracy. HMC obtains a significantly better FID score.

---

[2]https://github.com/openai/guided-diffusion

Table 3: Average accuracy, top-5 accuracy, and FID score for classifier-full guidance on ImageNet. The metrics are based on 50k generated samples for both sampling methods with score parameterizations.

|  | Sampler | Acc [%]↑ | Acc-5 [%]↑ | FID↓ |
|---|---|---|---|---|
| Score | Reverse | **50.0** | 83.9 | 14.5 |
|  | HMC-6C | 49.9 | **85.1** | **11.6** |

### 4.4 Image Tapestry

As our final experiment, we conduct an image tapestry experiment, similar to Du et al. (2023) and based on their code[3]. The goal is to generate a coherent image composed of spatially localized content, each region conditioned on different prompts. This task involves both classifier-free guidance and model composition—specifically, the combination of multiple overlapping text-to-image diffusion models, each responsible for a portion of the scene.

We use a pre-trained DeepFloyd-IF model[4] as the base diffusion model. To refine the generated samples, we apply Langevin dynamics with our MH-like correction. For each diffusion step ($T = 100$), we include 15 additional Langevin steps. The pseudo-energy difference is approximated via line integration using three additional evaluation points per step. We set the classifier-free guidance scale to $\lambda = 20.0$.

The resulting image is presented in Figure 2(a), which showcases the generated tapestry with different regions displaying distinct visual content. Figure 2(b) provides a schematic overview of the used prompts and their spatial layout. In total, nine content regions are specified: four located in the corners of the image, each with unique prompts, and five overlapping in the center, all guided by the same prompt to create a unified visual theme.

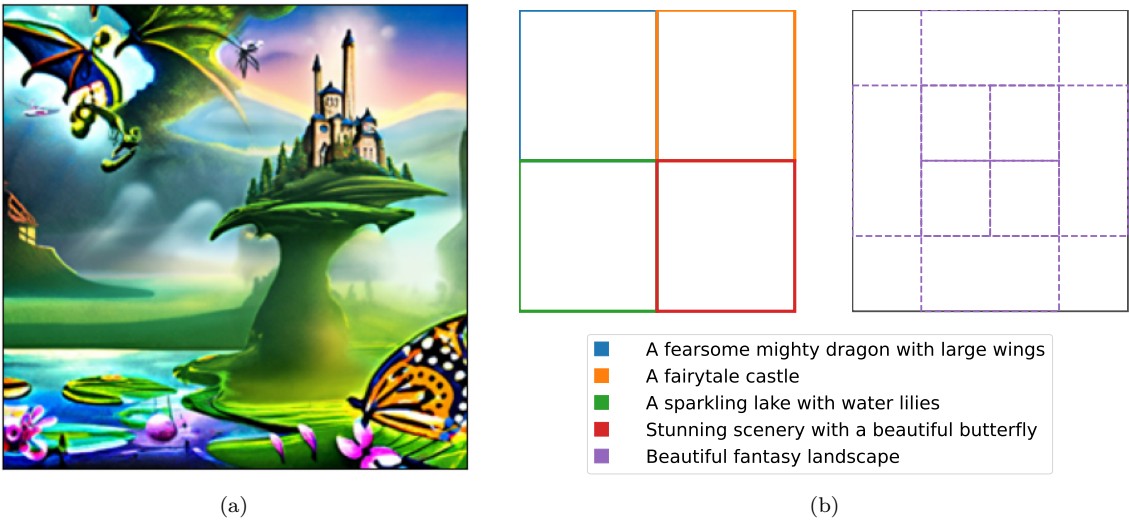

| | |
|---|---|
| (a) | (b) |

Figure 2: In (a), the generated tapestry image is shown with different content at various locations. In (b), the specified content and their positions are illustrated.

## 5 Discussion

The choice between score and energy parameterizations remains an intriguing and nuanced topic within diffusion-based generative modeling. In this work, we have provided additional empirical evidence suggesting that the score parameterization performs better in the standard reverse process.

---

[3]https://github.com/yilundu/reduce_reuse_recycle
[4]https://huggingface.co/DeepFloyd/IF-I-XL-v1.0

At the same time, we have shown that performance gains often attributed to the energy parameterization can, in fact, be recovered within a score-based framework. This is achieved by approximating pseudo-energy differences using a line integral of the model's noise predictions. Notably, this allows us to incorporate MH-like correction steps into a variety of MCMC samplers—without the need to explicitly train an energy-based model—yet still attain comparable improvements in sample quality.

A particularly interesting observation is that using a curve composed only of model evaluations from the HMC sampler appears to perform on par with using a straight-line path. This suggests that the proposed correction comes at virtually no additional computational cost in this case. However, it is worth noting that in higher-dimensional settings, additional intermediate points along the integration path may be required to maintain accuracy, which could increase the computational burden. This challenge might be addressed through more efficient numerical integration techniques, or by working in a lower-dimensional latent space, as is done in latent diffusion models. One persistent drawback of the energy parameterization is that it always requires an explicit gradient computation to recover the score function.

Another important consideration is computational cost. As we observed in the experiments, adding MCMC updates on top of the reverse diffusion process introduces a clear overhead in terms of additional score evaluations. Our approach is therefore not intended as a replacement for accelerated solvers, e.g., PF-ODE, which aim to minimize the number of function evaluations. Instead, we view MH-like corrections as an orthogonal contribution: they can be applied on top of any diffusion model and any sampler. In this sense, our method is complementary to existing work, targeting improved sample quality while remaining agnostic to the choice of backbone or solver.

In line with prior work such as Du et al. (2023), our experimental design deliberately focused on widely used baselines rather than the strongest available backbones. This choice allowed us to isolate the effect of the MH-like correction without confounding factors from architectural or solver improvements. While this comes at the cost of absolute FID values that are below the state of the art, the relative improvements we observe consistently demonstrate the benefit of our approach. We expect that applying our method to stronger architectures would yield proportionally similar gains, but leave this as an exciting direction for future work.

One limitation of the score parameterization is that the learned vector field is not guaranteed to be conservative, and therefore our line-integral construction does not in general yield an exact Metropolis–Hastings correction. Instead, it should be understood as a practically motivated procedure that is not theoretically exact. Importantly, our construction recovers an exact MH correction in the special case where the score is conservative, thereby aligning with the energy-based formulation. However, this reliance on score accuracy is not unique to our method: score-based approaches such as reverse SDE sampling, probability flow ODEs, or Schrödinger bridge formulations (Song et al., 2021; De Bortoli et al., 2021) also implicitly assume access to a correct score function. In practice, non-conservative approximations of the score nevertheless yield effective generative models. Recent work by Horvat & Pfister (2024) further supports this perspective by showing that strict conservativity is not required for score-based generative models to accurately represent data distributions. In the same spirit, our MH-like correction mechanism, though not grounded in exact conservativity, improves sample quality when applied to the reverse process. As an empirical sanity check, we also performed an experiment on a trained MNIST score model, where we compared pseudo-energy differences obtained from line and curve paths. The discrepancies were consistently very small (see Appendix B.1), suggesting that the score behaves approximately conservatively on the local scales relevant to MCMC proposals.

Still, the lack of theoretical guarantees may explain the slightly superior performance of the energy parameterization observed in the CIFAR-100 experiment. Developing better techniques for estimating pseudo-energy differences from score-based models—without requiring an explicitly trained energy function—thus remains a highly relevant and promising direction for future research.

## 6  Conclusion

We have introduced a method for extending the reverse diffusion process with MCMC sampling based on an MH-like correction step computed from the score function. This approach enables improved sampling for composed diffusion models without requiring an energy-based parameterization.

While previous work Du et al. (2023) demonstrated the benefits of MH correction under an energy parameterization, our method instead defines a pseudo-energy difference derived from the score, estimated via numerical integration. This allows us to apply MH-like corrections in the score-based setting—by far the most common in practice—and thereby make use of existing pre-trained diffusion models for composition tasks.

Our method can reuse intermediate evaluations from samplers such as HMC to compute the correction with little to no additional cost. In general, the accuracy of the MH-like correction depends on the numerical integration of the score, which may require more intermediate points as the dimensionality increases. While this can introduce some overhead, energy-based methods incur their own costs, such as differentiating the energy function. In practice, our corrected score-based samplers consistently match the performance of energy-based methods across a range of tasks, making them a practical alternative in settings where score-based models are already available.

Overall, our work extends the applicability of corrected MCMC sampling to the broad class of score-based diffusion models and opens the door to more flexible and modular composition of generative models.

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

# A  Experimental details

Here we provide more details about our different conducted experiments: Evaluating pseudo-energy differences, 2D composition, guided diffusion, and image tapestry.

The earlier experiments were conducted on a machine equipped with an NVIDIA GeForce RTX 3060, while the later experiments were run on a computing cluster with NVIDIA A100 Tensor Core GPUs.

## A.1  Evaluating Pseudo-Energy Difference

All models in this section were trained with the Adam optimizer using a learning rate of $10^{-3}$, together with a StepLR scheduler with step size 1 and decay factor 0.99.

**2D Gaussian:** We generated samples from a bivariate Gaussian distribution with mean $\boldsymbol{\mu} = (2,0)^\top$ and covariance $\boldsymbol{\Sigma} = 0.1I$, where $I$ is the identity matrix.

The diffusion models use $T = 100$ timesteps, with the noise schedule $\beta_t$ following the cosine schedule proposed in Nichol & Dhariwal (2021).

We use the same neural network architectures as the base for both the score and energy models. It is a residual network consisting of a linear layer (dim $2 \to 128$) followed by four blocks, and concluding with a linear layer (dim $128 \to 2$). Within each block, the input $x$ passes through a normalization layer, a SiLU activation, and a linear layer (dim $128 \to 256$). Subsequently, it is added with an embedded $t$ (dim 32) that has undergone a linear layer transformation (dim $32 \to 256$). The resulting sum passes through a SiLU activation and is further processed by a linear layer (dim $256 \to 256$). After that, another SiLU activation is applied, followed by a final linear layer (dim $256 \to 128$). The output of this linear layer is then added to the original input $x$ within the block. The embedding of $t$ is also learnable.

**MNIST:** The diffusion models use $T = 1000$ timesteps, with the noise schedule $\beta_t$ following the cosine schedule. For the score parameterization we trained a UNet-based architecture adapted to $28 \times 28$ grayscale images. The network uses a time embedding of dimension 112, implemented as sinusoidal position embeddings followed by two fully connected layers with GELU activations. The model begins with a $1 \times 1$ convolution mapping the input image to dimension 28, and proceeds through three down-sampling stages, a middle block, and three up-sampling stages. Each down/upsampling stage consists of two residual blocks with time conditioning, a linear or full attention layer, and either a strided convolution (downsampling) or nearest-neighbor upsampling with convolution (upsampling). The middle block contains two ResNet blocks and one full attention layer. Skip connections are applied between corresponding encoder and decoder layers, following the standard UNet design. The output stage concatenates the upsampled features with the initial projection and applies a residual block followed by a $1 \times 1$ convolution to map back to the image space.

For the energy parameterization, the architecture is identical, but the output head is replaced with an energy function whose gradient defines the score.

## A.2  2D composition

The composed distribution is defined by a product of two components, a Gaussian mixture and a uniform distribution with non-zero values on

$$\square = \{x \in \mathbb{R}^2 : -s_i \le x_i \le s_i, i = 1, 2\}, \tag{14}$$

where $s_1$ and $s_2$ are equal to 0.2 and 1.0, respectively. The eight modes of the Gaussian mixture are evenly distributed on a circle with a radius of 0.5 at the angles $\frac{\pi}{4}i$ for $i = 0, \dots, 7$, respectively. The covariance matrix at each mode is $0.03^2 \cdot I$, where $I$ is the identity matrix.

We use the same network architecture setup for score and energy as in the 2D Gaussian case (see Section A.1).

The metric log-likelihood is ill-defined as we may generate samples where the true distribution has no support (due to the uniform distribution). We address this problem by expanding the definition set of the uniform

distribution and redistributing one percent of the probability mass into this extended region. The whole set is defined as (14) except $s_1 = s_2 = 1.1$. Note that 99 percent probability mass remains inside the original definition set □.

The parameter $\beta_t$ follows the cosine schedule. For (U-)HMC, the damping coefficient is set to 0.5, the mass diagonal matrix has all diagonal elements equal to 1, and the stepsize for each $t$ is 0.03. For (U-)LA, the stepsize for each $t$ is set to 0.001.

## A.3 Guided diffusion for CIFAR-100

The parameter $\beta_t$ has a linear schedule as originally proposed in Ho et al. (2020). For (U)-HMC is the damping coefficient equal to 0.9 and the diagonal elements in the mass matrix are equal to $\beta_t$ for each $t$. The values of the stepsize parameters $a$ and $b$ were determined through a simple parameter search for the different MCMC methods and they can be found in Table 4. This was done for both the score and energy parameterizations, where the stepsize is defined as $\delta_t = a\beta_t^b$.

Table 4: The values of the stepsize parameters $a$ and $b$ obtained from a random parameter search for the different MCMC methods for both score and energy parameterization in the CIFAR-100 experiment, where the stepsize is defined as $\delta_t = a\beta_t^b$.

|  | MCMC | Stepsize Parameters | |
|---|---|---|---|
|  |  | a | b |
| Energy | U-LA | 9.22 | 1.40 |
|  | LA | 9.84 | 0.83 |
|  | U-HMC | 0.26 | 1.53 |
|  | HMC | 9.33 | 1.48 |
| Score | U-LA | 1.96 | 1.04 |
|  | LA | 9.84 | 0.83 |
|  | U-HMC | 0.26 | 1.53 |
|  | HMC | 4.03 | 1.34 |

To complement the quantitative results in the main text, we report in Table 5 the theoretical number of function evaluations (NFE) per generated sample for CIFAR-100. We separate forward passes (FP) and backward passes (BP), since energy parameterizations require BP for score evaluations whereas score parameterizations only require FP. Counts are computed per algorithm step and correspond to the maximum evaluations assuming all proposals are accepted in MH.

Table 5: Theoretical number of function evaluations (NFE) per generated sample on CIFAR-100. We report forward passes (FP) and backward passes (BP) separately. Counts are computed per algorithm step, independent of accept/reject outcomes.

|  | Sampler | NFE (FP) | NFE (BP) |
|---|---|---|---|
| Energy | Reverse | 0 | 1000 |
|  | U-LA | 0 | 6994 |
|  | LA | 6993 | 7993 |
|  | U-HMC | 0 | 6994 |
|  | HMC | 2997 | 7993 |
| Score | Reverse | 1000 | 0 |
|  | U-LA | 6994 | 0 |
|  | LA-8L | 55945 | 0 |
|  | U-HMC | 6994 | 0 |
|  | HMC-3C | 13982 | 0 |

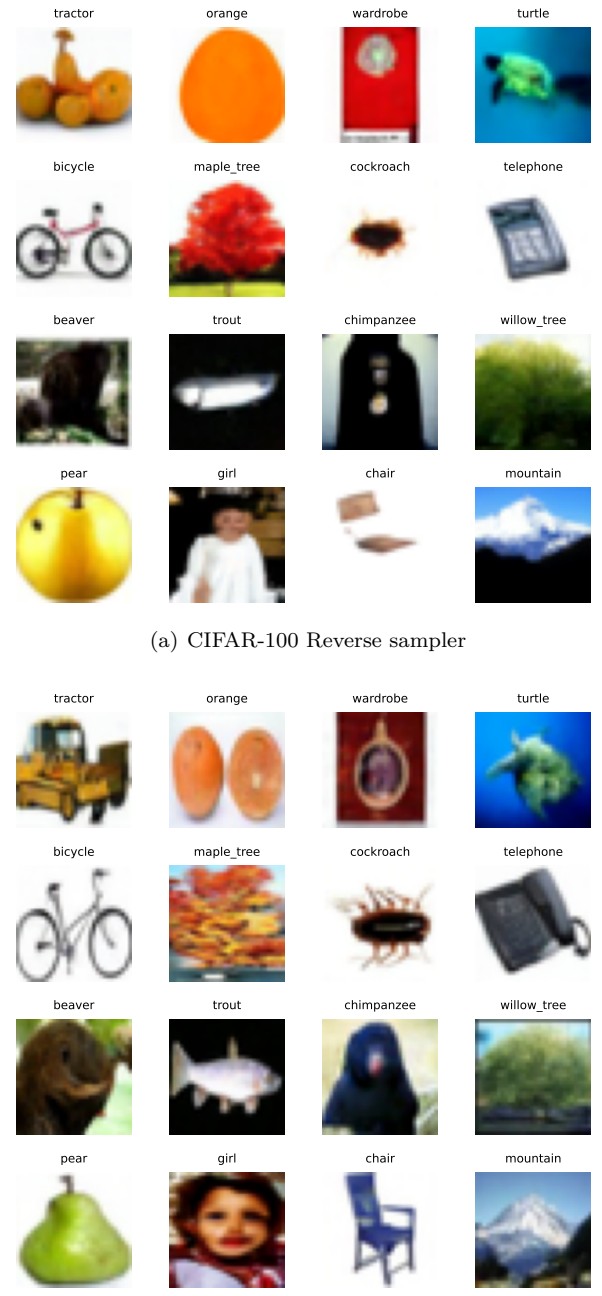

(a) CIFAR-100 Reverse sampler

(b) CIFAR-100 HMC sampler

Figure 3: Generated CIFAR-100 samples using (a) Reverse and (b) HMC.

To provide a qualitative comparison, we show representative generated samples in Figure 3. Each pair of images (a) and (b) is conditioned on the same class label and initialized from the same noise realization $x_T$, but generated with different samplers.

### A.4 Guided diffusion for ImageNet

Again, the parameter $\beta_t$ follows a linear schedule. The hyperparameters for the HMC include a damping coefficient set to 0.9, with the diagonal elements of the mass matrix being equal to $\beta_t$ for each $t$. The stepsize parameters for HMC, obtained from a simple parameter search, are $a = 1.87$ and $b = 1.51$.

The ImageNet dataset[5] used to compute the FID score is available for free to researchers for non-commercial use.

For the ImageNet experiment with classifier guidance, only score parameterizations were considered. Table 6 reports the corresponding NFEs per generated sample, again separating forward (FP) and backward (BP) passes. Here we compare the baseline reverse sampler with our MH-like corrected HMC variant (HMC-6C).

Table 6: Theoretical number of function evaluations (NFE) per generated sample on ImageNet. We report forward passes (FP) and backward passes (BP) separately. Counts are computed per algorithm step, independent of accept/reject outcomes.

|       | Sampler | NFE (FP) | NFE (BP) |
|-------|---------|----------|----------|
| Score | Reverse | 1000     | 0        |
|       | HMC-6C  | 19981    | 0        |

For ImageNet, representative samples are shown in Figure 4. As in the CIFAR-100 case, the pairs are conditioned on the same class label and share the same starting noise realization $x_T$, enabling a direct visual comparison between the samplers.

### A.5 Image tapestry

A cosine schedule is used for the parameter $\beta_t$. The stepsize parameters in this case is simply $a = 1$ and $b = 1$, i.e., $\delta_t = \beta_t$.

## B Additional experiments

### B.1 Sanity check: line vs. curve path on MNIST

We conducted an additional experiment on the MNIST dataset to further assess the effect of path choice when estimating pseudo-energy differences. The setup followed the same spirit as in Section A.1. Specifically, we trained 10 independent score models and sampled 2000 states from the forward process at various timesteps $t$. From each state, we performed a single HMC proposal consisting of 3 leapfrog steps with stepsize $\varepsilon_t = a\beta_t^b$, where $a = 4.03$ and $b = 1.34$ (as in the CIFAR-100 experiments). This produced a proposed state $\hat{x}_t$ for each sampled pair $(x_t, \hat{x}_t)$.

We then computed the pseudo-energy difference between $x_t$ and $\hat{x}_t$ using two different paths: (i) a straight line between the points, with $n = 10$ (Algorithm 2), and (ii) the curved path defined by the leapfrog steps themselves, with $m = 3$ (Algorithm 3).

The discrepancy between the two estimates was measured using the symmetric relative error

$$\frac{2|\Delta E_{\text{line}} - \Delta E_{\text{curve}}|}{|\Delta E_{\text{line}}| + |\Delta E_{\text{curve}}|}.$$

For each model, we computed the median relative error across the 2000 sampled pairs. We then report the mean and standard deviation across the 10 trained models, yielding a value of $0.022 \pm 0.002$. This small error indicates that line and curve integration paths produce very similar pseudo-energy differences, supporting the interpretation that the score behaves approximately conservatively on the local scales relevant for MCMC proposals.

---

[5]https://image-net.org/

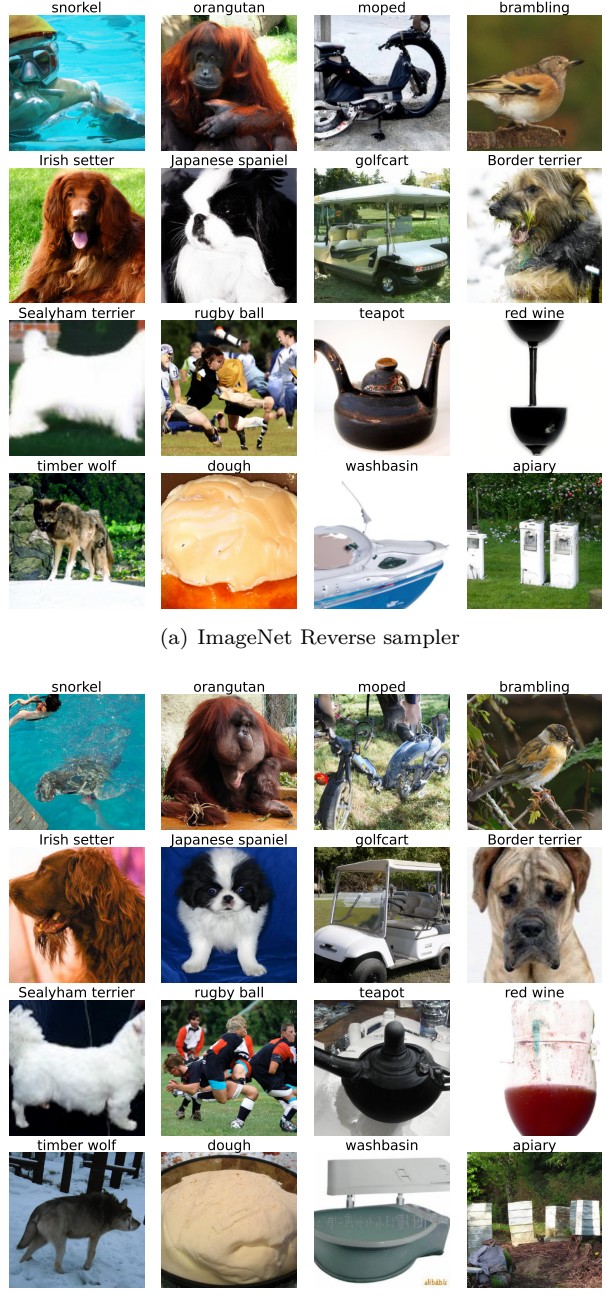

(a) ImageNet Reverse sampler

(b) ImageNet HMC sampler

Figure 4: Generated ImageNet samples using (a) Reverse and (b) HMC.

