# OpenReview forum: "MCMC-Correction of Score-Based Diffusion Models for Model Composition"
_TMLR — Rejected by TMLR_

### Review · Reviewer_1615 · 2025-07-21

**Summary Of Contributions:**

The paper proposes:

- **MH‐like correction for score‐based diffusion models**
  Introduces a Metropolis–Hastings–style acceptance rule (Eq. 11–12) , enabling MH‐corrected MCMC sampling without an explicit energy parameterization.
- **Two integration‐path algorithms**  (Straight‐line and curve path.  )
-  Experiments are as follows:

2D Gaussian & MNIST (§ 5.1), 2D composition task (§ 5.2, Table 1) .

CIFAR-100 and ImageNet (§ 5.3, Tables 2–3).

Image tapestry (§ 5.4, Figure 2).  speed and memory check (§ 5.2b, Table 1b).

**Audience:**

Yes

**Claims And Evidence:**

Yes

**Requested Changes:**

- 5.2 Table b, detail implementation specifics (framework, hardware) and include variance across runs to support “superior runtime performance” assertions.

- A question for clarification:
   Could there be direct pseudo‐energy treatment for mixture compositions beyond the sample‐then‐correct workaround in 4.2?

The rest, Please refer to weaknesses.

**Strengths And Weaknesses:**

### Strengths

- Builds on line‐integral interpretation of energy differences; recovers true MH when the score field is conservative.
- Practical efficiency: “Curve” integration reuses HMC evaluations to avoid extra model calls.

### Weaknesses

**Limited High-Dimensional Demonstrations**
   - All high-dimensional tests are constrained to pixel‐space diffusion on CIFAR-100/ImageNet. The method’s applicability to latent diffusion models (e.g., Stable Diffusion) is unexplored, leaving its performance on state-of-the-art generators uncertain. However, I believe this could be a minor issue.

**Classifier-Guidance Scale Not Justified**
 In Section 5.3 the classifier guidance weight λ is set to a fixed value (e.g., λ=20) with no ablation. It might be better for an ablation.

**Missing Pseudocode for Curve Integration**
It seems that Algorithm 1 describes the straight‐line integration path in detail; the curved‐path variant is only verbally sketched. It might be better to get full more detailed algorithm for curved-path.

---

> ### Author Response · Authors · 2025-08-27
>
> We thank the reviewer for the careful reading of our paper and for the constructive and encouraging feedback. We are glad that the reviewer highlighted the novelty of enabling MH-like corrections for score-based diffusion models, recognized the strengths of the line-integral formulation, and appreciated the practical efficiency of the curved-path strategy. We also thank the reviewer for raising several important points regarding clarity and scope, including high-dimensional demonstrations, classifier guidance, and the presentation of the curved-path algorithm. Below we respond to each point in turn.
>
> **Reviewer comment:**
> 5.2 Table b, detail implementation specifics (framework, hardware) and include variance across runs to support “superior runtime performance” assertions.
>
> **Response:**
> We thank the reviewer for this helpful suggestion. In the revised manuscript we have made this explicit by stating that the runtime measurements in Table 5.2b were conducted in JAX on an NVIDIA GeForce RTX 3060 GPU. In addition, we report the variance across multiple runs in Table 5.2b to support the runtime statements and ensure clarity and transparency.
>
> **Reviewer comment:**
> A question for clarification: Could there be direct pseudo-energy treatment for mixture compositions beyond the sample-then-correct workaround in 4.2?
>
> **Response:**
> We thank the reviewer for this interesting question. Our current sample-then-correct workaround strikes us as a natural way to handle mixture compositions when the goal is to obtain samples, especially in contrast to the approach of Du et al. (2023), which relies on the strong assumption of access to the relative normalizers. However, we are not aware of any alternative direct pseudo-energy treatment for mixtures.
>
> **Reviewer comment:**
> All high-dimensional tests are constrained to pixel-space diffusion on CIFAR-100/ImageNet. The method’s applicability to latent diffusion models (e.g., Stable Diffusion) is unexplored, leaving its performance on state-of-the-art generators uncertain. However, I believe this could be a minor issue.
>
> **Response:**
> It is correct that our quantitative high-dimensional experiments are conducted in pixel-space diffusion models (CIFAR-100 and ImageNet). Our focus in this work has been on establishing the principle that MH-like corrections can be applied to score-based diffusion models, rather than exhaustively covering all architectures. We note, however, that our tapestry experiment (§5.4) involves a latent representation, which provides an initial indication that the method is not restricted to pixel space. We see no conceptual barrier to extending our approach to latent diffusion models such as Stable Diffusion, and we view a more systematic investigation of this setting as an exciting direction for future work.
>
> **Reviewer comment:**
> In Section 5.3 the classifier guidance weight $\lambda$ is set to a fixed value (e.g., $\lambda=20$) with no ablation. It might be better for an ablation.
>
> **Response:**
> We have clarified in the revised manuscript that the choice $\lambda = 20$ follows a commonly used default in classifier-guided diffusion models and is consistent with the public implementation of Du et al. (2023). While an ablation over $\lambda$ could certainly provide further insight, our goal in this paper was to evaluate the effect of the MH-like correction under a representative and widely used setting rather than to tune the guidance strength.
>
> **Reviewer comment:**
> It seems that Algorithm 1 describes the straight-line integration path in detail; the curved-path variant is only verbally sketched. It might be better to get a more detailed algorithm for the curved path.
>
> **Response:**
> We thank the reviewer for this helpful suggestion. We agree that the curved-path variant was previously described only at a high level, and that a more explicit algorithmic description improves clarity. In the revised manuscript, we have clarified this by presenting the straight-line and curved-path variants in separate algorithms. We believe this substantially improves the clarity and presentation of our method.
>
> ---
>
> **Reference:**
> Du, Y., Durkan, C., Strudel, R., Tenenbaum, J. B., Dieleman, S., Fergus, R., Sohl-Dickstein, J., Doucet, A., & Grathwohl, W. S. (2023). *Reduce, reuse, recycle: Compositional generation with energy-based diffusion models and MCMC*. In International Conference on Machine Learning (ICML), pp. 8489–8510. PMLR.

---

### Review · Reviewer_zgyY · 2025-07-26

**Summary Of Contributions:**

This paper proposes a novel Metropolis–Hastings (MH) correction scheme for score-based diffusion models, which typically lack an explicit energy formulation and therefore cannot leverage traditional MH-based sampling. The key idea is to approximate the density ratio required in the MH acceptance probability using a line integral of the score function between proposed and current samples. Since the score function is not generally conservative, the method explores specific path choices for the line integral, such as linear interpolations and paths informed by Hamiltonian Monte Carlo (HMC) trajectories. The authors empirically validate the method on synthetic and real-world data.

**Audience:**

Yes

**Broader Impact Concerns:**

None.

**Claims And Evidence:**

No

**Requested Changes:**

Provide a more detailed justification for the use of line integrals to approximate log-density ratios. If the method is to be taken seriously as a rigorous MH correction, the implications of non-conservativeness must be addressed more thoroughly. As it stands, the proposed method is simply a heuristic and ultimately detracts from the rigor of the field.

The use of “product distribution” is inappropriate here, as its standard usage refers to the law of a random variable with independent components.

Include a more thorough discussion of the computational cost associated with the proposed line integral calculation. In particular, provide computational comparisons to baseline sampling methods.

Correct the use of \citep and \citet as appropriate, e.g., \citep should be used in the second sentence of the introduction.

Remove the extraneous comma in Eq. (13) before $ds$.

**Strengths And Weaknesses:**

Strengths:

The paper is well-written and addresses an important challenge in the field—leveraging MH corrections in the widely-used score-based diffusion models without requiring reparameterization to energy-based forms. The paper includes empirical studies on both synthetic and real datasets, which showcase the potential utility of the proposed method. In particular, the approach is appealing in practice as it allows the use of existing pre-trained score models, which are plentiful and well-optimized.

Weaknesses:

The most significant limitation is the absence of theoretical grounding for the proposed MH correction. As the authors acknowledge, the score function is path-dependent due to non-conservativeness, so the line integral does not correspond to a proper log-density difference in general. This undermines the principled justification for using MH acceptance rules and raises concerns about the correctness and interpretability of the method.

Since the line integral depends on the chosen path, it is unclear what the acceptance probability is actually measuring or correcting for, weakening the conceptual rigor of the method.

The computational overhead of computing the line integrals—especially the number of neural network evaluations required—is not sufficiently discussed. The experiments should also include a computational comparison for fairness. This is a critical aspect when considering the method's practicality and scalability.

---

> ### Author Response · Authors · 2025-08-27
>
> We thank the reviewer for the thoughtful and constructive feedback. We are glad that the reviewer found the paper well-written, identified the importance of enabling MH corrections for score-based diffusion models, and appreciated the empirical evaluation across synthetic and real datasets. We also appreciate the reviewer’s concerns, which help us clarify both the conceptual foundations and the practical implications of our work. Below we respond to each point in detail.
>
> **Reviewer comment:**
> The most significant limitation is the absence of theoretical grounding for the proposed MH correction. As the authors acknowledge, the score function is path-dependent due to non-conservativeness, so the line integral does not correspond to a proper log-density difference in general. This undermines the principled justification for using MH acceptance rules and raises concerns about the correctness and interpretability of the method.
>
> **Response:**
> We thank the reviewer for this important comment. We agree that line integrals over score functions do not in general yield exact log-density ratios, due to the lack of conservativeness. For this reason, we have been careful in our terminology throughout the paper, referring to our approach as a “pseudo-energy” method and describing the resulting procedure as an “MH-like correction” rather than a theoretically exact MH step.
>
> Our intention is not to overstate the theoretical foundations of the method, but rather to offer a practically motivated heuristic that leverages the score in situations where the underlying density is unknown or intractable. This is in line with common practices in score-based generative modeling, where many methods rely on approximate log-density surrogates (e.g., reverse SDE, probability flow ODE, and Schrödinger bridge techniques). Importantly, in the special case of energy-based models—where the score is exactly the gradient of the log density—our method recovers a proper MH correction, thereby providing a consistency check and formal grounding in that idealized setting.
>
> We further conducted an experiment on a trained MNIST score model, where we compared the pseudo-energy difference obtained from a straight line path and from an HMC curved path between proposed and current states. We found that the discrepancy between the two was very small across many sampled proposals, which supports the intuition that the score is approximately conservative on the local scales relevant for MCMC steps. This also aligns with our main experiments, where both line and curve choices yielded similar sample quality. We have included this experiment in the Appendix.
>
> Finally, we have updated the manuscript with clarifications and an expanded discussion of this limitation.
>
> **Reviewer comment:**
> The use of “product distribution” is inappropriate here, as its standard usage refers to the law of a random variable with independent components.
>
> **Response:**
> We agree that the term “product distribution” is not appropriate in this context. We have revised the manuscript to consistently use “product composition,” which more accurately reflects our intended meaning.
>
> **Reviewer comment:**
> Include a more thorough discussion of the computational cost associated with the proposed line integral calculation. In particular, provide computational comparisons to baseline sampling methods.
>
> **Response:**
> We fully agree that the computational aspects of our method should be made clearer. In the revised manuscript we describe the number of MCMC steps and the additional function evaluations required for the MH correction, and we emphasize more explicitly that this makes our approach computationally heavier than baseline diffusion samplers, since we add MCMC updates at every reverse step (similar in spirit to predictor–corrector schemes in, e.g., Song et al., 2021). For transparency, we have also added a table reporting the number of forward passes and backward passes required by score and energy models with and without MH correction, compared to baseline methods.
>
> Finally, we have expanded the discussion section to include a dedicated remark on computational cost.
>
> **Reviewer comment:**
> Correct the use of citep and citet as appropriate, e.g., citep should be used in the second sentence of the introduction. Remove the extraneous comma in Eq. (13).
>
> **Response:**
> We thank the reviewer for pointing out these typos. We have corrected the citation style and removed the extraneous comma in the revised version.
>
> ---
>
> **Reference:**
> Song, Y., Sohl-Dickstein, J., Kingma, D. P., Kumar, A., Ermon, S., & Poole, B. (2021). *Score-based generative modeling through stochastic differential equations*. In International Conference on Learning Representations (ICLR).

---

### Review · Reviewer_PQSp · 2025-07-30

**Summary Of Contributions:**

The paper considers adding an MH-correction step to MCMC-based sampling from diffusion models. The MH step conventionally requires access to the marginal density evaluations, while this paper shows how it can be approximated using only score information. The main contribution is calculating a proposal density ratio using a line-integral of the denoiser between them, which is a great idea.

**Audience:**

No

**Broader Impact Concerns:**

No issues

**Claims And Evidence:**

No

**Requested Changes:**

Major
- The paper needs to present experiments where the baselines achieve close to SOTA performance; and then show the improvements of MH-MCMC against MCMC samplers consistently in all experiments. Poor FIDs imply model training issues, and this can cause or mask sampling issues. For instance, the nvidia's edm/edm2 models could be used.
- The paper also should compare to existing samplers of PF-ODE, Euler/Heun/etc. The results should also compare number of function evaluations.
- The experiments needs to be more transparent wrt the sampling hyperparameters and other metrics. The paper needs to convince the reader these are appropriately chosen for each sampler. Some visualisations of the samplings should be shown for explicitness.

Minor
- Sec 2.3. states that EBMs are commonly trained by DSM. There are no citations or justifications about this. My impression is that EBMs are commonly trained by contrastive divergence, not by DSM. This requires some clarifications: I think the paper is here not talking about EBM’s in general [as in Lecun’06], but just EBM/diffusion hybrid models, without making this explicit.
- I can’t follow where eq 3 comes from. This is perhaps borderline obvious stuff, but there is no citation or derivation here.
- It would be useful for readability to include a background section about ULA and UHMC to cover the basics. Right now eq (3)..(x)..(4) [numbering all equations would be convenient for the reader] jumps right in the middle of ULA with little explanation on what ULA is or where in ULA this kernel goes, or what a kernel is (ULA Is often presented without kernels).
- Tau is undefined
- ULA is typically presented without kernels using just \nabla E [eg. wikipedia]. The paper needs to explain the kernelisation.
- The citations have some issues: I don’t think Neal’96 mentions ULA or UHMC.
- Conventionally we would sample with PF-ODE or perhaps with reverse SDE. How does the MCMC approach relate to them, and what advatanges or disadvantages do they have over them? Why do we want to do MCMC instead of eg. PF-ODE? Naively it seems that the MCMC is much more expensive, while commonly we would try to instead have diffusion samplers with very few function evaluations (eg. less than 10). Can you discuss?
- Fig1 seems to show that all methods failed. Isn’t this strange? Can you explain?
- Why did you use classifier guidance in CIFAR-100? Isn’t CFG preferred?
- What are all the different samplers in the tables? These need to be explicit.

**Strengths And Weaknesses:**

S: Evaluating density ratios through line integrals is a strong contribution

S: The idea of MH-MCMC has lots of potential

S: The results are promising

W: The experiments are inconsistent and show multiple issues. The 2D experiment seems to have failed throughout. The CIFAR experiment achieves weak FIDs showing training problems. ImageNet also shows weak FIDs, and is missing most of the samplers. The experiments are throughout missing sampling baselines (different stochastic samplers, PF-ODE), are missing NFEs and runtimes.

W: I'm a bit confused what is the main promise of using MCMC in the first place. Diffusion literature has optimised sampling speed and commonly reports SOTA sampling results on just a handful of function evaluations. The MCMC approach likely requires orders of magnitude more, and I'm not seeing the motivation. Currently I'm a bit uncertain of the "audience" requirement of TMLR: why are these results useful or interesting to some part of the domain?

---

> ### Author Response · Authors · 2025-08-27
>
> We thank the reviewer for recognizing the novelty of our MH-like correction via line integrals and for acknowledging the potential of MCMC approaches in the context of diffusion models. We also appreciate the reviewer’s detailed feedback, which helps us clarify both the scope and the presentation of our work. In particular, we acknowledge the reviewer’s concerns regarding the strength of the experimental results on 2D, CIFAR-100, and ImageNet. Our focus in this paper was not to pursue state-of-the-art FID scores, but rather to isolate and evaluate the effect of the proposed MH-like correction in a controlled and transparent way. We have revised the manuscript to make this positioning clearer, and to emphasize that our method is complementary to stronger backbones and solvers. Below we address each of the reviewer’s points in turn.
>
> ---
>
> #### Major
>
> **Reviewer comment:**
> The paper needs to present experiments where the baselines achieve close to SOTA performance; and then show the improvements of MH-MCMC against MCMC samplers consistently in all experiments. Poor FIDs imply model training issues, and this can cause or mask sampling issues. For instance, the nvidia's edm/edm2 models could be used.
>
> **Response:**
> We thank the reviewer for this important comment. Our goal in this paper was not to pursue state-of-the-art FID values, but rather to isolate and evaluate the effect of the proposed MH-like correction across different settings. For CIFAR-100 we followed the widely used training configurations from Ho et al. (2020), and for ImageNet we relied on the pretrained OpenAI guided-diffusion model. These choices deliberately mirror prior work such as Du et al. (2023), which similarly prioritized methodological clarity and comparability over absolute performance. While these backbones are not the most recent or strongest available, they are widely adopted in the literature and thus provide a clear and transparent baseline for assessing the relative impact of our method.
>
> Our results consistently show that the MH-like correction improves sample quality relative to the corresponding uncorrected samplers, even when the absolute FID values are not competitive with the latest architectures. We emphasize that our contribution is complementary: the proposed correction is model-agnostic and can, in principle, be combined with any diffusion backbone. To avoid confusion, we have clarified this scope explicitly in the discussion section.
>
> **Reviewer comment:**
> The paper also should compare to existing samplers of PF-ODE, Euler/Heun/etc. The results should also compare number of function evaluations.
>
> **Response:**
> We appreciate this suggestion. Our experiments already reported the number of function evaluations (NFEs) implicitly, but in the revised manuscript we have made this explicit by presenting the number of forward passes and backward passes directly in the experimental tables (in Appendix).
>
> Regarding baselines, we deliberately focused on the standard reverse diffusion process rather than on a broad set of improved solvers, in order to cleanly isolate the effect of the proposed MH-like correction. This choice allows us to demonstrate our core contribution—that the correction consistently improves sample quality irrespective of the underlying sampler—in the clearest possible way. Similarly to Du et al. (2023), we prioritize clarity and comparability over absolute performance. We now make this rationale explicit in the text, and also clarify that our correction scheme is agnostic to the underlying sampler and could in principle be combined with solvers such as PF-ODE or Euler/Heun.
>
> Finally, in the discussion section we have emphasized that, while much of the diffusion literature has focused on accelerating sampling (reducing NFEs), our contribution is complementary: we target improvements in sample quality through a correction mechanism.
>
> **Reviewer comment:**
> The experiments need to be more transparent wrt the sampling hyperparameters and other metrics. The paper needs to convince the reader these are appropriately chosen for each sampler. Some visualisations of the samplings should be shown for explicitness.
>
> **Response:**
> We followed the hyperparameter choices from Du et al. (2023) wherever possible, and otherwise relied on standard settings from the literature. In the revised manuscript we have made this more transparent by explicitly stating the hyperparameter sources and values both in the results section and in Appendix.
>
> To ensure reproducibility, we also provided the full codebase at submission time, which will be made publicly available upon acceptance.
>
> Finally, to complement the quantitative metrics, we have added representative visualizations of generated samples in the appendix. These are obtained by conditioning on the same class and initial state, to make the effect of our MH-like correction directly comparable to the baseline.

---

> > ### Author Response · Authors · 2025-08-27
> >
> > #### Minor
> >
> > **Reviewer comment:**
> > Sec 2.3. states that EBMs are commonly trained by DSM. There are no citations or justifications about this. My impression is that EBMs are commonly trained by contrastive divergence, not by DSM.
> >
> > **Response:**
> > Our intention was not to claim that DSM is the predominant training method for EBMs in general, but rather to highlight that DSM provides one way of viewing diffusion models in relation to EBMs. In the revised manuscript we have therefore clarified the text to:
> > “In the diffusion setting, one approach to training EBMs is denoising score matching (DSM).”
> > This avoids ambiguity and makes explicit that our statement concerns the diffusion context rather than EBMs broadly.
> >
> > **Reviewer comment:**
> > I can’t follow where eq (3) comes from. This is perhaps borderline obvious stuff, but there is no citation or derivation here.
> >
> > **Response:**
> > We have clarified in the revised manuscript that equation (3) arises from identifying the noise prediction model $\epsilon_\theta(x_t,t)$ with the score of an energy function, i.e. as an EBM. This makes explicit the connection that establishes the equivalence between denoising score matching and the diffusion training loss (up to a factor of $\sigma_t^2$) (Song et al., 2021).
> >
> > **Reviewer comment:**
> > It would be useful for readability to include a background section about ULA and UHMC to cover the basics. Right now eq (3)..(x)..(4) jumps right in the middle of ULA with little explanation on what ULA is or where in ULA this kernel goes, or what a kernel is (ULA Is often presented without kernels).
> >
> > **Response:**
> > We agree with the reviewer that, since MCMC plays a significant role in our paper, it is beneficial to provide more theoretical background on this topic. In the revised manuscript we have therefore expanded the section to include a clearer introduction to both LA and HMC. Specifically, we now:
> > - Introduce $\tau$ to explicitly denote the MCMC iteration index, in contrast to the diffusion step $t$.
> > - Define the transition kernel and explain how it is used for LA and HMC.
> > - Present the LA kernel explicitly as a Gaussian update based on the score.
> > - Describe the HMC kernel by introducing momenta, leapfrog integration, and how proposals are generated.
> > - Explain why these methods are called “unadjusted” and how adding an MH correction recovers LA and HMC, respectively.
> >
> > This revision also addresses the reviewer’s comment that $\tau$ was previously undefined.
> >
> > We believe this makes the section substantially clearer and more self-contained, while still keeping the exposition concise.
> >
> > **Reviewer comment:**
> > ULA is typically presented without kernels using just $\nabla E$ (e.g. Wikipedia). The paper needs to explain the kernelisation.
> >
> > **Response:**
> > Indeed, ULA is most often presented directly through its update rule involving $\nabla E(x)$ rather than in terms of a kernel. In our setting, however, the kernelised view is natural: it places LA and HMC on the same footing and connects directly to the MH formulation, where the transition kernel appears in the acceptance probability. This also keeps the presentation consistent with the rest of our paper, where kernels play a central role in describing MCMC moves. We have clarified this in the revised manuscript.
> >
> > **Reviewer comment:**
> > The citations have some issues: I don’t think Neal’96 mentions ULA or UHMC.
> >
> > **Response:**
> > We thank the reviewer for this careful observation. Neal (1996) was originally cited as a classical reference on Hamiltonian Monte Carlo, where the need for a Metropolis–Hastings correction is emphasized. While this indirectly relates to the unadjusted variant, we agree that it is clearer to use a direct reference for U-HMC, and we have updated the citation accordingly.

---

> > > ### Author Response · Authors · 2025-08-27
> > >
> > > **Reviewer comment:**
> > > Conventionally we would sample with PF-ODE or perhaps with reverse SDE. How does the MCMC approach relate to them, and what advantages or disadvantages do they have over them? Why do we want to do MCMC instead of e.g. PF-ODE? Naively it seems that the MCMC is much more expensive, while commonly we would try to instead have diffusion samplers with very few function evaluations (e.g. less than 10). Can you discuss?
> > >
> > > **Response:**
> > > Conventional samplers such as probability flow ODE and reverse SDE focus on directly discretizing the diffusion dynamics. These approaches are typically optimized for efficiency, often requiring only a small number of function evaluations, but they may introduce discretization bias.
> > >
> > > By contrast, the MCMC approach we explore can (at least in theory) guarantee correct sampling even with finite step sizes when combined with a Metropolis–Hastings correction. Of course, in practice we rely on an approximate score model, which limits this guarantee, but the correction still provides a mechanism to reduce bias. Another advantage is that MCMC can be flexibly combined with other samplers as a “corrector” step, similar in spirit to predictor–corrector schemes already used in diffusion models (e.g., Song et al., 2021).
> > >
> > > The main drawback of MCMC is computational cost: it requires multiple score evaluations per diffusion step, making it less efficient than PF-ODE or other accelerated samplers. Our work is motivated by the observation that, despite this cost, adding MCMC steps can improve sample quality, making it a complementary strategy to existing discretization-based approaches rather than a replacement. We have clarified this trade-off explicitly in the revised discussion section.
> > >
> > > **Reviewer comment:**
> > > Fig1 seems to show that all methods failed. Isn’t this strange? Can you explain?
> > >
> > > **Response:**
> > > The apparent failure in Figure 1 is in fact expected: as shown by Du et al. (2023), simply adding score functions does not correspond to a valid product composition, so we should not expect this approach to succeed. Annealed MCMC is specifically motivated by this limitation: in annealing, the intermediate distributions are treated as design choices that guide the chain toward the final target, while preserving asymptotic correctness. We have clarified this explanation in the revised manuscript.
> > >
> > > **Reviewer comment:**
> > > Why did you use classifier guidance in CIFAR-100? Isn’t CFG preferred?
> > >
> > > **Response:**
> > > We chose classifier guidance for CIFAR-100 primarily to align with the setting used in Du et al. (2023), on which our work is built. Our aim is not to optimize for state-of-the-art sampling performance, but to isolate and study the effect of the proposed MH correction. Importantly, our approach is agnostic to the specific guidance mechanism: it can be applied both with classifier guidance and with classifier-free guidance (as we demonstrate on Tapestry).
> > >
> > > **Reviewer comment:**
> > > What are all the different samplers in the tables? These need to be explicit.
> > >
> > > **Response:**
> > > We have now clarified what the different samplers in the tables correspond to. This has been achieved partly by expanding the theoretical background on MCMC samplers, and more importantly by adding Algorithms 2 and 3, which specify in detail how the line- and curve-based variants are implemented. These additions make clear which variants are used in the experiments. Together, these changes substantially clarify the experimental setup and the meaning of each table entry.
> > >
> > > ---
> > >
> > > **References**
> > >
> > > - Ho, J., Jain, A., & Abbeel, P. (2020). *Denoising diffusion probabilistic models*. Advances in Neural Information Processing Systems (NeurIPS), 33, 6840–6851.
> > > - Du, Y., Durkan, C., Strudel, R., Tenenbaum, J. B., Dieleman, S., Fergus, R., Sohl-Dickstein, J., Doucet, A., & Grathwohl, W. S. (2023). *Reduce, reuse, recycle: Compositional generation with energy-based diffusion models and MCMC*. In International Conference on Machine Learning (ICML), pp. 8489–8510. PMLR.
> > > - Song, Y., Sohl-Dickstein, J., Kingma, D. P., Kumar, A., Ermon, S., & Poole, B. (2021). *Score-based generative modeling through stochastic differential equations*. In International Conference on Learning Representations (ICLR).
> > > - Neal, R. M. (1996). *Bayesian Learning for Neural Networks*. Springer, New York. (Lecture Notes in Statistics, Vol. 118).

---

### Decision · Action_Editor_Sopp · 2025-09-13

**Recommendation:** Reject

**Additional Comments:**

See above for a detailed discussion.

**Audience:**

Yes

**Audience Explanation:**

See above.

**Claims And Evidence:**

No

**Claims Explanation:**

I have read both the paper, the reviews and the rebuttal discussions.

I agree that the problem tackled by the authors is meaningful and important. There is interest in the community for such methods. However, as pointed out by the reviewers the paper suffers from several flaws which I think need to be addressed before considering publication in TMLR.

First of all, the empirical results are of poor quality. The Imagenet results in particular are quite underwhelming. Most baselines (available in open source methods) would reach a FID of around 2. I do not see any meaningful gain in performance here. While I know that this is not the main point of the authors I think that when dealing with generative models it is important to be able to show that the proposed empirical method does not degrade the performance. In my opinion, there is not enough convincing evidence that it is the case here.

As a result I find that claims such as "Comparing the score and energy parameterizations, their performances share similar characteristics. Interestingly, the reverse process favors the score parameterization, supporting the claim that this less restricted
approach better models the score function. However, the energy parameterization sees larger improvements
from the added MCMC steps." are not met with suitable evidence. Hence, to be more precise, I think that there is not enough empirical work to justify that "Through experiments on synthetic and real-world data, we show that our MH-like samplers offer comparable improvements to those obtained with energy-based models".

Second, poor empirical results would be more acceptable if there were strong theoretical results associated with the paper. This is not the case here as emphasized by reviewer zgyY: "The most significant limitation is the absence of theoretical grounding for the proposed MH correction. As the authors acknowledge, the score function is path-dependent due to non-conservativeness, so the line integral does not correspond to a proper log-density difference in general. This undermines the principled justification for using MH acceptance rules and raises concerns about the correctness and interpretability of the method."

As a result I will recommend the rejection of the paper.

**Resubmission Of Major Revision:**

The authors may consider submitting a major revision at a later time.